# Assessment of Pain Treatments in Disorders of Upper Limbs: A Qualitative Study Protocol Based on Patients' Experiences

Weronika Maria Karcz [1,2,†], Eva Artigues-Barberà [3,4,5,*], Marta Ortega Bravo [3,6,7], Alejandra Pooler Perea [2,8,†], Jose María Palacín Peruga [2,8] and Iraida Gimeno Pi [2,4,*]

1 Bordeta Magraners Primary Care Center, Gerència Territorial Lleida, Catalan Health Institute (ICS), Carrer Boqué, s/n, 25001 Lleida, Spain

2 Fundació Institut Universitari per a la Recerca a l'Atenció Primària de Salut Jordi Gol i Gurina (IDIAPJGol), Gran Via Corts Catalanes, 587, 08007 Barcelona, Spain

3 Research Support Unit Lleida, Fundació Institut Universitari per a la Recerca a l'Atenció Primària de Salut Jordi Gol i Gurina (IDIAPJGol), Gran via de les Corts Catalanes, 587 àtic, 08007 Barcelona, Spain

4 Balafia Primary Care Center, Gerència Territorial Lleida, Catalan Health Institute (ICS), Av de Rosa Parks, s/n, 25005 Lleida, Spain

5 Department of Nursing and Physiotherapy, Faculty of Nursing and Physiotherapy, University of Lleida, Carrer de Montserrat Roig, 2, 25198 Lleida, Spain

6 Cappont Primary Care Center, Gerència Territorial Lleida, Catalan Health Institute (ICS), 1 d'Octubre, 1, 25001 Lleida, Spain

7 Department of Medicine, Faculty of Medicine, University of Lleida, Carrer de Montserrat Roig, 2, 25008 Lleida, Spain

8 Onze de Setembre Primary Care Center, Gerència Territorial Lleida, Catalan Health Institute (ICS), Av. Onze de Setembre, Passeig de l'Onze de Setembre, 10, 25005 Lleida, Spain

* Correspondence: eartigues.lleida.ics@gencat.cat (E.A.-B.); igimeno.lleida.ics@gencat.cat (I.G.P.)

† These authors contributed equally to this work.

**Abstract:** Chronic musculoskeletal pain (CMP) is one of the most common symptoms of musculoskeletal disorders. Carpal tunnel syndrome (CTS) and subacromial syndrome (SAS) are the most prevalent musculoskeletal disorders of the upper limbs. By collecting the opinions of patients with CTS and SAS, we aim to identify variables that could be introduced in the follow-up of CMP, and to detect barriers and facilitators of its treatments to improve their acceptance. This qualitative study is being conducted in Lleida, Spain, and explores the experiences and feelings of patients, and their acceptance of the standard of care. It follows the consolidated criteria for reporting qualitative research (COREQ) through focus groups, addressing issues with rigor and representativeness. By collecting patients' opinions, we expect to obtain valuable information to complement the set of variables previously used by health professionals in the follow-up of CMP, and to understand treatment barriers and facilitators.

**Keywords:** qualitative; life experiences; nursing; upper limb; focus groups

## 1. Introduction

In 1979, the International Association for the Study of Pain (IASP) defined pain as "An unpleasant sensory and emotional experience associated with actual or potential tissue damage, or described in terms of such damage" [1]. Such definition has become globally accepted by healthcare professionals and researchers in the field of pain, and has been adopted by several professional, governmental, and nongovernmental organizations, including the World Health Organization (WHO). Although subsequent revisions and updates have been made to the list of associated pain terms (1986, 1994, 2011), the IASP definition has remained unchanged [2] and describes pain as subjective [1].

According to a systematic review, approximately 1710 million people in the world experience musculoskeletal disorders, with chronic musculoskeletal pain (CMP) being

one of the most common symptoms [3]. In 2005, 10% to 40% of the general population of Spain presented musculoskeletal disorders and CMP, and this percentage was higher in women and elderly [4]. Additionally, in Spain, CMP is the main reason for chronic pain consultation in Primary Care (PC) [5]. Previous studies found that CMP represents a significant burden for the individual and society [6]: it negatively influences the physical and mental health of patients, who enter a vicious circle of pain, social isolation, depression, and inactivity [7]. In particular, carpal tunnel syndrome (CTS) and subacromial syndrome (SAS) are among the most prevalent causes of CMP in the upper limbs [5]. A systematic review of 32 clinical trials reported that 3.8% to 4.9% of the world's population has CTS, and that the frequency is higher in women aged 50 to 59 years [8]. Another systematic review indicated that the prevalence of SAS reaches 9.2% in patients under 20 and 62% in patients over 80 years of age, in different countries [9].

The majority of studies on CMP favor a conservative treatment, which, in addition to having fewer complications than surgery, has shown a functional long-lasting improvement [10]. The first therapeutic approach is oral non-steroidal anti-inflammatory drugs for 5–7 days, together with rest [11]. In cases of non-response, the next step consists of rehabilitative therapy or local injections with glucocorticoids [11]. In addition, several studies showed that ultrasound-guided injections of glucocorticoids produce significant clinical improvements, in comparison to "blind" injections [12,13].

However, pain treatments could be differently seen by each professional involved in therapy, and by each patient. As shown by Kohrt et al., incorporating patients' experiences, social determinants, and comorbidities into healthcare models would help professionals to provide personalized care [14]. The latter would focus on the characteristics of each patient to better adapt therapeutic and preventive measures [14]. Finally, the contribution of patients during the follow-up of disorders allows healthcare professionals to learn about patients' experiences and perspectives, improving the determination of the efficiency of different treatments, and encourages personalized care [15,16].

In this context, our objective is to identify which variables could be introduced in the follow-up of CMP, and to detect the barriers and facilitators of treatments, in order to improve their acceptance. We will achieve that by collecting the opinions of patients with CTS and SAS. Our research is being conducted in the context of a regular care practice for people with CTS or SAS treated in PC Health Centers, within the framework of a study with mixed methodology (qualitative and quantitative), and following the recommendations of the Medical Research Council. Our multidisciplinary approach with the collaboration of patients could better address society's needs and guide public health policies towards more experiential population health [17,18], with the research team better understanding patients' historical, cultural, and social backgrounds.

## 2. Methods

### 2.1. Study Design, Materials, and Equipment

This is a qualitative study with a phenomenological perspective that began in Lleida, Spain, in February 2022, and is expected to end in December 2023. It follows the consolidated criteria for reporting qualitative research (COREQ) [17] and addresses the issues rigorously, with representation and reflexivity.

We are collecting information on individual experiences of pain from upper limb disorders (CTS and SAS) among focus groups of patients. This way, we will gain insight into patients' opinions on follow-up methods and current treatments. We will mean to obtain information about three thematic blocks, and dimensions of interest, structures according to the researching multidisciplinary team opinion. The approach allows holistic configuration study, subject to a flexible and dynamic structure that let to address deeply the livelihoods, opinions and experiences of the participants.

## 2.2. Detailed Procedure

The sample is intentional [17], and participants are selected through sampling based on pragmatic and convenience criteria [19]. The participants are adults with CTS or SAS from the PC Health Center of Lleida, inside the Institut Català de la Salut (ICS), which in turn is part of the Public Health System. Inclusion criteria are patients diagnosed with CTS or SAS; women and men over 18 years of age or older; who voluntarily signed informed consent for inclusion in the study; who had or had not previously received surgical treatment of the area. Exclusion criteria are patients who have cognitive deterioration. In addition, inclusion criteria for focal groups are considered both according to age and sex. Candidates who do not have a fluent and coherent speech are excluded. Women's, men's and mixed groups are organized.

Patients are accessed through medical professionals from different Lleida PC Health Centers and pre-selected according to the CTS or SAS clinical records, obtained from the electronic medical records stored in the centralized ECAP database. Professionals are in charge of making the first approach, asking patients if they are interested in participating, giving them the informed consent and the study information sheet, and collecting their contact details. Then, patients are given time to consider the request before the research team calls them to confirm the profile, request their informed consent, and finally arrange their participation in a focus group. Groups are defined according to profiles and adapted to the patients' availability and preferences.

Six focus groups of 8–10 people will be formed with a semi-structured script (Table 1). These groups will identify beliefs, opinions, experiences, feelings, sensations, and perceptions on the follow-up and treatments for CMP, such as infiltration (I), ultrasound-guided infiltration (UI), and oral pharmacological treatment (PT). Meetings will be held in an available room of the PC Health Center, which will have to be easily accessible to participants, intimate, and comfortable for a fluent communication with no interruptions, interferences, or noises. Additionally, it should allow for a circular distribution of furniture. Two researchers will gather the necessary information: one will lead the group and the other will supervise the participants' interactions and collect all the data. The approximate duration of the meeting will be 1 h and a half. It will be recorded through the Teams tool from the Microsoft Office 365 program, linked to the corporative cloud of the ICS, and stored in OneDrive. Only the main investigator and the research team members will be granted access by means of an ICS institutional email account. The maximum discursive plurality will be sought in terms of sex, age, and sociocultural status, to represent the whole population [19]. A sociodemographic survey will be carried out among all the participants. The number of focus groups could vary until information saturation. Before the meeting, both oral and written consent will be obtained by the participants and kept under lock and key in a drawer in the research team's institutional office. Besides, participants will be compelled to keep the confidentiality.

## 2.3. Ethical Considerations

This project has been evaluated and accepted by the Research Ethics Committee of the Primary Care Research Institute IDIAPJ Gol with the code 4R22/067. The study is being carried out following the recommendations of the Declaration of Helsinki and Tokyo. The records of the discussion groups, the data, and the variables collected will be treated anonymously, guaranteeing confidentiality. The evaluation will be done in compliance with Regulation 2016/679 of the European Parliament and of the Council, which took place on 27 April 2016, regarding the protection of natural persons, the processing of personal data, and the free circulation of these data. The evaluation will also comply with the Organic Law 3/2018, which took place on 5 December, on the Protection of Personal Data, and the Guarantee of Digital Rights. The participant lists and the group recordings will be kept electronically and protected by utilizing identification codes with dissociation of the data. The files will be located in a corporate environment with restricted access, remain in the research team's possession, and will be treated confidentially. The data will undergo an anonymization process prior to their use. Anonymized database will be accessible and only

used by the research and collaborating teams working on this study. Data will be stored until the results are disseminated. If patients give consent for transfer and portability for other scientific research purposes, their wishes will be respected. Data will be preserved for 10 years at least. Data linked to a publication or verification of the investigation will be preserved in the long term. The data will be stored in an ICS institutional repository. Personal data collected for this research will not be shared with other databases.

**Table 1.** Semi-structured script for focal interviews.

| Topics/Thematic Blocks | Content to Investigate during the Interviews |
|---|---|
| Experiences and opinions on follow-up and treatments for CMP in CTS or SAS | Information and experience: What do patients know about CTS and/or SAS and the pain relief using I, UI, and PT treatments? What is their experience? Repercussions: How do these syndromes affect patients daily? How do they affect their quality of life, sleep/rest, and emotions? Coping: How do patients deal with these disorders, and what coping strategies do they use or have heard about? Follow-up: According to patients, which professional should do the follow-up and how? What tools or resources do patients need from healthcare professionals? |
| Barriers and facilitators of treatments and their acceptance | Barriers: According to patients, what would the disadvantages of the pain relief treatments be? Facilitators: According to patients, what would the advantages of the pain relief treatments be? Acceptance: What pain relief treatments could cause fear, uncertainty, distrust, insecurity, or other negative emotions in patients? Expectations and results achieved with the received treatment. Expected changes and satisfaction with the treatments and the received follow-up. |
| Professional support | Support: What are the patients' experiences and expectations regarding the support given and the monitoring of treatments by professionals? Needs: What are the patients' needs regarding the help from healthcare professionals? Beliefs about the known treatments, anticipation or acceptance and follow-up. |

*2.4. Data Management and Analysis*

An inductive thematic content analysis will identify the core meanings within a wide and varied content and thus obtain a unique and original analysis of the previously defined thematic blocks. [20] (Table 1).

Specifically, we will perform verbatim transcripts of the recorded sessions, properly anonymized. Such transcripts will be then entered into the Atlas-Ti software and examined, coded and categorized by thematic units/dimensions. The outcome will be analyzed by the research team. To give more rigor to the project and validity to the study, and to ensure the quality of the interpretation, the data will be triangulated between researchers. Afterwards, a thematic analysis of the literal transcription of the interviews will be carried out manually and separately by several research team members. Finally, a joint effort will unify the results by reconciling the differences, and a summary will be generated. An analysis of the discourse will be carried out following the model proposed by Braun and Clarke [21]. If necessary to ensure the quality of the interpretation of the results of the study, the collaboration of experts in anthropology or other disciplines will be requested.

## 3. Expected Results

With this study, we expect to obtain information about patients' experiences and opinions on the follow-up and treatments of CMP in CTS or SAS. This way, we would identify new variables to be introduced in the follow-up of CMP, e.g., articular functionality, pain range, participation in the treatment process, complementary tests, use of concurrent treatments, complications of the standard of care, specific complications of the puncture treatments (I and UI), and patient satisfaction. Additionally, we will obtain information on treatment (I, UI, and PT) barriers and facilitators that patients may encounter, and determine treatment acceptance.

Being able to work with the most accepted follow-up model represents a solid basis to establish the therapeutic alliance between health professionals and patients, and to address psychosocial influences on pain. According to Cuyul et al., interventions on the biological dimension should be congruent with interventions on psychosocial factors [22]. In line with this, the interventions of our multidisciplinary team should not only seek biological changes, but should also encourage a change in the person's perception of reality, including the environment and their psychological condition, which play a major role in recovery [22].

Possibly our results will be in line with findings from other publications: injected treatments are considered less favorably than oral medications [23]; patients usually feel misunderstood and abandoned by the healthcare professionals [24]; the structural conditions, appreciation of the benefit, estimation of risk and time, and the patients' attitude may have an impact on treatments [25]. Moreover, treatments could have a different impact depending on sex, individuals' sensitivity, and pain expression. This has been deeply analyzed by different investigators who concluded that sex affects all the steps of the pain pathway: from signaling to perception, expression, and treatment [26,27].

This research may open new possibilities through a better investigative approach, involving common people as active participants and improving patient empowerment [28]. Citizen collaboration strengthens the applicability to the clinical practice that distinguishes investigation in PC. Finally, citizen collaboration guarantees a better social and cultural sensitivity and improves viability, external validity, and cost-effectiveness of the research [28].

These contributions will be integrated as new approaches in clinical practice and will be evaluated in a randomized clinical trial setting that will be done in a later phase. The objective of the trial will be to evaluate the functional outcomes of I, UI, and PT treatments, cost-effectiveness and cost-utility, and overall patient satisfaction. The study is registered on ClinicalTrials.gov under code 4R22/067.

## 4. Discussion

Pain is a subjective sensation that is experienced by individuals throughout their life, affecting their well-being [2,29]. It is influenced by biological, psychological, and social factors [2], as well as by political and cultural situations, and the economic status of the country [30,31]. Additionally, the relationship of pain with age [32], and gender [30], is well known. Finally, pain can have physical causes, but can also be induced by emotions [33], and people that are anxious and under psychological stress may feel greater physical pain [30].

Sometimes individual sensations of patients experiencing pain are unknown to the professionals taking care of them that, therefore, do not offer the best pain management. Indeed, healthcare providers may equip patients with guidelines for self-management and lifestyle, but patients' actions are influenced by multiple factors, including job requirements, beliefs, and environment [34]. Thus, the control of health lies with and is in the hands of patients rather than professionals [35].

The management of patients with pain can be complex and may require a multidisciplinary approach considering all the factors described above to provide safe and quality care [34]. Moreover, to alleviate their condition, sometimes patients may seek alternatives (e.g., physiotherapy, which has been shown to be effective [10,36], and coping strategies that may influence the monitoring of pain. Patients with shoulder pain, for example, reported different barriers and facilitators, such as: support, knowledge, time/daily routine, access

to equipment, beliefs, expectations, motivation, therapeutic response, and influence of the clinician [37]. Foo et al. highlighted the importance of considering the different views of professionals and users in policy formulation and planning for community care [34]. From there, the proposal was developed to include patient experiences in the study to improve treatments for different disorders. For example, Nielsen et al. explored the experiences of patients with a functional motor disorder to improve clinical services and their outcomes [24]. Additionally, Skogö Nyvang et al. described the experiences of patients who underwent knee replacement and determined whether the expectations of the surgery were met [38].

In the present study, we propose to integrate the information provided by patients with CMP caused by CTS and SAS in their follow-up [39]. Knowing how patients live and considering their needs will help professionals to improve care practice and service provision, and focus on personalized medicine [14]. Finally, a social approach to pain, involving individuals, communities, institutions, and decision-makers from different settings, is considered crucial to improve and support the management of multiple chronic conditions [34].

*Limitations*

Our study design has some limitations. First, incorporating the focal group technique within investigation could make the participants feel uncomfortable because of the group presence or the dominance of one member. However, this could be solved through the correct leadership of the focal group conductor, either by means of his experience or his skillfulness may imply its success. Besides, the aim of the investigation, the sensitivity of the treated subject, and any other logistical aspects could also solve this issue. Another limitation is represented by the data analysis and transference, which can be quite difficult when dealing with a subjective matter such as pain. A good design of the study with the right adaptability and the flexibility to investigate new aspects will be needed.

## 5. Expected Conclusions

We expect to understand for the first time the experience of patients, and so to incorporate it in the follow-up strategies and acceptance of the treatments, acknowledging the difficulties and facilitations they may find.

**Author Contributions:** E.A.-B., I.G.P. and J.M.P.P. contributed to the study concept and research design. W.M.K. and A.P.P. selected and reviewed the literature. W.M.K., E.A.-B., M.O.B., A.P.P., J.M.P.P. and I.G.P. participated in the writing and critical review of the manuscript and approved the final version. All authors have read and agreed to the published version of the manuscript.

**Funding:** No specific grants from any funding agencies in the public, commercial, or non-profit sectors were received for this study.

**Institutional Review Board Statement:** The study was conducted in accordance with the Declaration of Helsinki, and approved by the Research Ethics Committee of the Primary Care Research Institute IDIAPJ Gol with the code 4R22/067.

**Informed Consent Statement:** The informed consent of all study participants will be obtained.

**Acknowledgments:** A sincere thank you to all healthcare professionals for helping in the recruitment phase of the study and to the patients.

**Conflicts of Interest:** The authors declare no conflict of interest.

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
