# Peer review of "Assessment of Pain Treatments in Disorders of Upper Limbs: A Qualitative Study Protocol Based on Patients’ Experiences"

_nursrep, doi:10.3390/nursrep13020070_

Round 1

Reviewer 1 Report

REVIEW FOR MANUSCRIPT ENTITLED “ASSESSMENT OF PAIN TREATMENTS IN DISORDERS OF UPPER LIMBS: A QUALITATIVE STUDY PROTOCOL BASED ON PATIENTS’ EXPERIENCES”.

Thank you for providing me with the opportunity to review the manuscript entitled “Assessment of pain treatments in disorders of upper limbs: A qualitative study protocol based on patients’ experiences”.

Congratulations for the work done. It is important to know the experience of patients with carpal tunnel syndrome (CTS) and subacromial syndrome (SAS).

There is an important research work but there are certain issues that should be modified.

.

1.       Introduction:

In the introduction, there is much discussion even with other types of patients and other pathologies. This should be included in a "discussion" section that does not appear as such in the publication.

For example, this paragraph does not contribute anything interesting to the publication. Although qualitative studies are done on other pathologies or dysfunctions, it is not necessary for this introductory section:

“Different works in the past have used qualitative methodology to study the experiences of the treatments for different disorders. For example, Nielsen et al. explored the experiences of  patients with a functional motor disorder to improve the clinical services and their outcomes [16]. Moreover, Skogö Nyvang et al. described the experiences of patients who underwent knee replacement and determined whether the expectations of the surgery were met [17]. Finally, Turner et al. studied the experiences of adolescents about different treatments for type 2 diabetes to improve treatment concordance [18].”

2.       Methods

The methodological explanation of this qualitative study is crearly: sampling strategy has been followed for the recruitment of participants; purposive sampling; data saturation, reflexivity…

It is necessary to specify whether there were any patient inclusion or exclusion criteria in the recruitment process. The only criterion specified was the inclusion in the study of patients with CTS or SAS.

3.       Discussion

It would be necessary to create a discussion section to justify why this study has been carried out and the comparison with the rest of the scientific literature.

4.       Limitations

This section should go at the end, after the discussion section and before the conclusion section.

5.       Conclusion

Despite the fact that no data were obtained, an objective of the study protocol was created: “The objective of this study is to explore the opinions of patients diagnosed with CTS and SAS to identify which variables could be introduced in the follow-up and the acceptance of treatments, as well as to detect the barriers and facilitators of the same”.

Despite the fact that no data have been obtained, an objective of the study protocol has been created. According to this, it is necessary to put a section of conclusions where it is specified what is expected. For example, according to the objective, it is expected to know the experience of the patients, which is not known so far, and through this to create strategies in the follow-up and acceptance of the treatments.

Author Response

Dear Reviewer 1, 

We attached our reponses.

Sincerely

Reviewer 2 Report

First of all, congratulations on your work proposal. I look forward to hearing the results of the intervention. As an aspect of improvement I would encourage you to include in the introduction as well as in the future dissertation references to the conservative treatment of physiotherapy in this pathology. I am enclosing a reference of interest for your research.

Klokkari, D., & Mamais, I. (2018). Effectiveness of surgical versus conservative treatment for carpal tunnel syndrome: A systematic review, meta-analysis and qualitative analysis. Hong Kong Physiotherapy Journal38(02), 91-114.

I congratulate you on your work

Author Response

Dear Reviewer 2, 

We attached our reponses.

Sincerely

Reviewer 3 Report

This manuscript by Karcz et al. describes a qualitative study protocol for assessing response to treatment of pain in disorders of the upper limbs. The stated goal is to determine variables that impact treatment for disorders such as carpal tunnel syndrome and subacromial syndrome. The proposed work could uncover valuable information, but the manuscript would benefit from increased clarity and detailed methodology.

Major comments:

1)      Careful proofreading by a native English speaker would significantly improve the overall quality of the manuscript.

2)      Given the importance of sex on pain perceptions, it would be useful to specify ideal sex distribution of participants

3)      Additional information about consent should be included- will it be obtained via virtual or written means? How will forms be sent and stored?

4)      Inclusion criteria should be more carefully defined- will all participants be 18+? How will diagnosis be confirmed? Will patients be comorbidities be included? Will patients have undergone infiltration (I), ultrasound-guided infiltration (IU), and/or oral pharmacological treatment (PT) prior to participation?

Minor comments:

1)      Ln 44-45 “variables exposed by the patients” is unclear

2)      Ln 55-56 It is unclear why there are dashes present mid-word

3)      Ln 95-99 The meaning here is unclear, but quantitative research is mentioned while the proposed study is qualitative

4)      Ln 230-231 The meaning here is unclear

5)      Ln 142 Which Microsoft tool will be used for recording?

Author Response

Dear Reviewer 3, 

We attached our reponses.

Sincerely

Round 2

Reviewer 1 Report

REVIEW FOR MANUSCRIPT ENTITLED “ASSESSMENT OF PAIN TREATMENTS IN DISORDERS OF UPPER LIMBS: A QUALITATIVE STUDY PROTOCOL BASED ON PATIENTS’ EXPERIENCES”.

Thank you for providing me with the opportunity to review the manuscript entitled “Assessment of pain treatments in disorders of upper limbs: A qualitative study protocol based on patients’ experiences”.

First of all, thank you for having made all the changes proposed by the reviewers. The work done is to be appreciated.

However, I will now specify certain aspects that still need to be modified.

1.       Introduction:

The proposed modifications have been perfectly implemented.

2.       Methods

The inclusion and exclusion criteria are poor and have not been properly performed. IT IS NECESSARY TO SPECIFY MORE CONCRETELY. 

As inclusion criteria it would be necessary to include, for example, patients diagnosed with CTS or SAS; women and men over 18 years of age; who voluntarily signed informed consent for inclusion in the study; who had not previously received surgical treatment of the area; who did not have cognitive deterioration.

3.       Discussion

I agree with your justification but one could briefly discuss, as you did in the introduction above, the rationale for conducting the study. Justify it with other scientific publications already published.

4.       Limitations

It is perfect.

5.       Conclusion

The modification is correct.

In this regard, the following scientific paper is suitable for publication after minor revision.

Author Response

Dear Reviewer 1,

We attached our responses.

Sincerely.

Reviewer 3 Report

I would recommend professional proofreading of the manuscript be performed.

Author Response

Dear Reviewer 3,

We attached our responses.

Sincerely.
